# A Novel Traffic Prediction Method Using Machine Learning for Energy Efficiency in Service Provider Networks

**DOI:** 10.3390/s23114997

**Published:** 2023-05-23

**Authors:** Francisco Rau, Ismael Soto, David Zabala-Blanco, Cesar Azurdia-Meza, Muhammad Ijaz, Sunday Ekpo, Sebastian Gutierrez

**Affiliations:** 1CIMTT, Department of Electrical Engineering, Universidad de Santiago de Chile, Santiago 9170124, Chile; 2Department of Computer Science and Industry, Universidad Católica del Maule, Talca 3480112, Chile; dzabala@ucm.cl; 3Department of Electrical Engineering, Universidad de Chile, Santiago 8370451, Chile; cazurdia@ing.uchile.cl; 4Department of Engineering, Faculty of Science and Engineering, Manchester Metropolitan University, Manchester M1 5GD, UK; m.ijaz@mmu.ac.uk (M.I.); s.ekpo@mmu.ac.uk (S.E.); 5Faculty of Engineering, Universidad Autónoma de Chile, Santiago 7500912, Chile; sebastian.gutierrezl@usach.cl

**Keywords:** energy efficiency, machine learning, telecom services operator, traffic prediction

## Abstract

This paper presents a systematic approach for solving complex prediction problems with a focus on energy efficiency. The approach involves using neural networks, specifically recurrent and sequential networks, as the main tool for prediction. In order to test the methodology, a case study was conducted in the telecommunications industry to address the problem of energy efficiency in data centers. The case study involved comparing four recurrent and sequential neural networks, including recurrent neural networks (RNNs), long short-term memory (LSTM), gated recurrent units (GRUs), and online sequential extreme learning machine (OS-ELM), to determine the best network in terms of prediction accuracy and computational time. The results show that OS-ELM outperformed the other networks in both accuracy and computational efficiency. The simulation was applied to real traffic data and showed potential energy savings of up to 12.2% in a single day. This highlights the importance of energy efficiency and the potential for the methodology to be applied to other industries. The methodology can be further developed as technology and data continue to advance, making it a promising solution for a wide range of prediction problems.

## 1. Introduction

The increasing demand for energy in contemporary industry, global warming, and the development of new communication technologies such as the Internet of Things (IOT), 5G, and B5G have necessitated research into energy-saving strategies in the telecommunications sector, especially among telecommunications service operators (TSOs). From 2010 to 2018, the global energy demand for data centers increased from 194 TWh to 205 TWh, according to research [1]. According to [2], by the year 2030, telecommunications networks will consume up to 51% of the world’s electricity if their energy efficiency is not significantly improved. Consequently, energy efficiency is becoming essential for existing and future 5G and beyond networks.

Some TSOs have designed networks with redundant links to avoid congestion in high-availability schemes (active/passive configuration) and load balancing (active/active configuration) [3]. From the perspective of energy efficiency, these designs can be considered as inefficient energy expenditure, because these links are always active [4]. The studies in [5,6] show that links are underutilized by only 40% in the central network, commonly called core network (CN), and that during off-peak hours, it is much lower. Moreover, there is not much difference in energy consumption between equipment at full load and standby mode [5]. To increase bandwidth capacity, TSOs connect the routers through multiple physical cables that form a single grouped logical link. The grouped logical links are also called link aggregation groups (LAG) or bundle Ethernet (BE). Link aggregation in the Ethernet was standardized in the IEEE 802.3ad standard and later renamed as IEEE 802.1ax to maintain consistency with other 802.3 standards [7].

The current CN of a TSO is composed of multiple networks, and trunk lines are often formed by many grouped optical fibers to increase capacity and add resilience. Connections with the number of sublinks within a BE or LAG range from 2 to approximately 20 in a typical TSO [6]. Additionally, there is the problem of the energy consumption of these links, which is even more when they are grouped. In recent years, several works have been published and proposed to reduce the energy consumption of simple links (unaggregated) [8]. One of the standards that helps solve this problem is IEEE 802.3az, which specifies Ethernet energy efficiency (EEE), which is a method for reducing the energy used by an Ethernet device during low link utilization periods [9]. The premise of EEE is that Ethernet links have idle time and therefore the opportunity to save energy during that period of time. The method is called low power idle (LPI), but it is only for copper interfaces [10]. Regarding LAGs, several energy efficiency studies have been carried out using different optimization methods [6]. Additionally, there are works that propose threshold point algorithms, as in [11], and others [12] that work with software-defined network (SDN) controllers. Moreover, ref. [13] has used predictive techniques to reduce LAG energy, using a simple moving average (SMA).

On the other hand, neural networks have begun to be used to predict time series, because time series are the optimal method to describe network traffic behavior, in particular regarding recurrent neural networks (RNNs) and their variants, long short-term memory (LSTM) and gated recurrent unit (GRU) [14,15,16,17]. There is also a neural network that stands out for its speed called online sequential extreme learning machine (OS-ELM), which has been shown to be efficient and especially fast compared with gradient-based networks [18,19,20]. This paper’s primary objective is to predict traffic in a short period of time to activate and deactivate the ports of the link aggregation (LAG or BE) between two nodes within a TSO network in order to save energy on fiber optic links while maintaining a high quality of service (QoS) for clients; the main contributions of this work are presented below:1.A novel method is proposed and developed to compare different types of neural networks in terms of their ability to process time series data, specifically in real-time traffic analysis. This methodology aimed to evaluate the performance of various neural network models and identify the most suitable option for the task.2.A novel bundle Ethernet energy efficiency methodology was designed. This algorithm was based on the expected traffic and used the best-performing neural network, selected by the methodology outlined in point 1.3.The proposed traffic prediction method and energy-saving Ethernet bundle methodology were evaluated. The performance of the traffic prediction methodology was compared between neural networks. The energy-saving Ethernet bundle was evaluated in terms of energy savings by comparing the performance algorithms proposed in point 2. The results were analyzed to determine the feasibility and effectiveness of the proposed solution.

The remainder of this paper is structured as follows: Section 2 presents related works in LAG or BE energy efficiency, as well as the most popular machine learning models and neural networks for traffic prediction. Section 3 details the methodology involved in the selection of a neural network. Section 4 outlines the stages used in the methodology for the development of two energy-efficient algorithms. Section 5 describes the use case, including network topology, traffic description, and network equipment specifications. In Section 6, the results of training a neural network using training and testing data, along with various evaluation metrics, are presented for the use case. In Section 7, the performance of the proposed energy efficiency optimization algorithms is compared with the base case, and the results are presented for the use case. Finally, relevant conclusions and future works are presented in Section 8.

## 2. Related Works

In this Section, we review the related works on traffic prediction based on neural networks and energy efficiency in grouped links (LAG or BE).

### 2.1. Works Related to Methods of Traffic Prediction Based on Machine Learning and Neural Networks

Analyzing historical traffic is a critical challenge for generating an accurate model that reflects the structure of the series in order to allow for prediction and classification of future events [21]. Time series consist of a succession of ordered numerical data points. The problem of time series prediction is the forecast of future activity from past values and the related patterns [21,22].

For traffic prediction, several time series forecasting techniques can be used, grouped into two types of linear and nonlinear methods. Nonlinear methods are more suitable for predicting traffic due to the existing noise and the complex nature that traffic presents [23]. Neural networks are being widely used to predict time series [24].

According to the study in [25], convolutional neural networks (CNNs) and RNNs are the most widely used deep learning models for short-term traffic prediction. CNNs are good at capturing spatial characteristics, and RNNs are good at capturing the temporal characteristics of traffic data. Within deep learning and recurrent networks, there are variations such as the LSTM and GRU neural networks. Currently, predicting traffic with neural networks has made the analysis of time series an essential part of data modeling in a wide range of industries, including finance, health, transportation, and the environment [26,27,28,29,30,31].

In [22], the authors used a GRU neural network, a variant of LSTM, to predict the traffic flow of ships within an area of wind parks. The results of GRU were compared with the autoregressive integrated moving average (ARIMA) model, as well as support vector machine (SVM) and LSTM deep learning models, with GRU being the winner.

In the logistics sector, the prediction of passenger flow in metro stations has been carried out by [32], as well as the use of parking modeling in [33] and urbanization planning based on vehicle traffic prediction [34]. In the health field, the prediction of driver stress and drowsiness for accident prevention [35] and the prediction of the monthly progression of Alzheimer’s disease [36] have also been demonstrated. With regard to the telecommunications sector, the prediction of TSO network traffic load has been used to avoid overloads, minimize response time, and optimize resource use [15,19,37,38].

With the arrival of the 5G standard, traffic prediction will be more difficult due to its heterogeneous nature. The coexistence of different networks and significantly different characteristics make traffic prediction, management, and optimization a difficult task. Therefore, the use and adaptation of neural networks will increasingly be used to analyze and manage network traffic based on data [39]. The diverse use of machine learning models in telecommunications operators and neural network research have led to more and more comparisons between them.

In [40], the use of cellular network traffic prediction for dynamic resource optimization in wireless backhaul networks is discussed. In [14], the authors state that the LSTM neural network is a type of recurrent neural network architecture, which is trained with a gradient-based learning algorithm. The framework proposed by the authors utilizes real traces from a TIER-1 TSO. With these traces, predictions were made at different time spans and compared with a combination of the classic method, such as ARIMA, and RNN. The results obtained in comparison with the ARIMA and RNN models show that the LSTM model performs well with a low normalized RMSE for the entire dataset and also generates predictions at very short time scales (less than thirty seconds). In the work in [41], the RNN, LSTM, GRU, deep neural network (DNN), and bidirectional LSTM (BLSTM) networks were compared, and the result was that LSTM was one of the networks that performed the best in terms of traffic prediction, due to having lower error metrics. In [42], the authors show the prediction of internet traffic in Telecom Italia, comparing deep learning models to conventional machine learning models. The performance of LSTM, GRU, and two conventional machine learning architectures, random forest (RF) and decision tree (DT), was compared for the prediction of mobile Internet traffic. The predictive quality of the models was evaluated using the root mean squared error (RMSE) and mean absolute error (MAE). Both deep learning algorithms were effective in modeling Internet activity and seasonality, both within days and over 2 months. The deep learning models outperformed the conventional machine learning models, placing the LSTM network as the winner over the GRU network in the experiments.

In the study in [15], it is shown that 5G networks can face network traffic peaks due to their numerous connections, so they focus on predicting these traffic peaks through deep learning techniques such as RNN, LSTM, and GRU from a real network. In terms of prediction, LSTM and GRU outperform RNN by 4.98% and 4.56%, respectively, and in terms of computational complexity, GRU is the worst compared with RNN and LSTM by 12.16% and 0.13%, respectively. Finally, since the times between GRU and LSTM are similar, when seeking greater precision for traffic peaks, the LSTM model performed the best. In [43], the prediction performance of the recurrent models RNN, LSTM, and GRU was compared with nonrecurrent models, such as XGBoost and RF, LASSO linear regression, and prediction models based on moving averages. The results indicate that RNN and its variants outperformed the other methods. The best nonrecurrent model was XGBoost. In comparison with XGBoost, GRU and RNN reduced 15% in the RMSE metric and 8% in MAPE. In another study, by [44], recurrent neural networks and their variants are compared with traditional statistical methods, such as ARIMA, seasonal ARIMA (SARIMA), and CNN. Three different congestion scenarios are tested: full day, morning peak hour, and afternoon peak hour. The results indicate that for the RMSE metric, the LSTM neural network had performances of 5.8, 7.9, and 10.2 and LSTM of 6.7, 8.6, and 10.9, depending on the scenario tested.

As previously mentioned, RNN, LSTM, and GRU have gained ground in traffic prediction compared with machine learning models (XGBoost, RF, SVM, and DT), conventional methods such as LASSO linear regression, ARIMA, and SARIMA, and other CNNs. One of the factors that hinders the RNN, LSTM, and GRU deep learning networks from being faster in prediction is the factor of seeking their optimization based on minimizing the gradient; thus, alternatives to this, without losing predictive performance, have been sought.

One of the neural networks that has attracted interest in recent times is the extreme learning machine (ELM) network, due to its good results in prediction and extremely fast training algorithm [45]. ELM is a particular type of feed-forward neural network. The learning mechanism allows for significantly faster training speed compared with classic neural networks in a variety of scenarios. A variation of this neural network is called online sequential extreme learning machine (OS-ELM). The advantage of OS-ELM over ELM is that it allows the algorithm to learn sequential data online, part by part, using the recursive least squares method [46]. In the study by [19], the authors compare the LSTM recurrent neural network with the OS-ELM neural network. The results show that OS-ELM outperforms LSTM in terms of computational cost by a factor of 2300, which is extremely high, and in terms of network prediction, OS-ELM was similar to LSTM. The OS-ELM neural network is simpler in architecture than LSTM, which makes it faster in terms of complexity than recurrent RNNs and their variants. The latter neural networks may have better accuracy in certain cases, but it depends on the input data, the architecture of the time series, and the hyperparameters that are configured.

For prediction models, the goal of evaluation metrics is to minimize error. For regression and prediction models, MAE and RMSE metrics are most commonly used to evaluate the performance of the model. For example, in studies such as [15,19,37,47,48,49,50], the RMSE metric has been used, and in others such as [36,37,47,51], the MAE metric has been used. The mean absolute percentage error (MAPE) is also used in some cases to evaluate regression and prediction [52,53]. RMSE and MAE are metrics that are dependent on the scale. MAPE is a percentage error metric. The RMSE metric is very sensitive to outliers, while the impact of these is reduced with MAE. On the other hand, MAE cannot indicate the bias of predictions in terms of overfitting or underfitting. MAPE can differentiate this type of bias and imposes a penalty on predicted values above the real ones.

In this study, the highest possible accuracy in prediction is required, with the processing time being relaxed to the maximum extent possible to the time of obtaining system data. As this depends on the data of the time series, this work makes a comparison between OS-ELM and recurrent neural networks and their variants (RNN, LSTM, and GRU) to determine which of these networks is better in terms of accuracy (RMSE, MAE, and MAPE) and computational time.

### 2.2. Works Related to Energy Efficiency in Link Aggregation Groups or Bundle Ethernet

In the field of energy-efficient link aggregation, several methods and techniques have been proposed to reduce the energy consumption of LAGs or BEs in telecommunications networks, which can be divided into four categories: optimization methods, threshold points, SDN controllers, and predictive techniques.

In [6], an integer linear programming (ILP) formulation was proposed to optimize the energy of LAGs. The energy savings were significant, reducing energy consumption by 79% compared with a normal network. Another study, in [54], investigated energy reduction in clustered links using a mixed-integer linear programming (MILP) model, showing a 50% reduction compared with shortest-path routing. In [55], the use of link aggregation activation and deactivation was studied using linear programming (LP) optimization, and it was shown how to find the most energy-efficient link configuration for all links of a BE. Energy savings ranged between 10% and 30% depending on the chosen configuration. In [56], convex optimization was used with the water-filling algorithm, reducing energy consumption by up to 50% in EEE links, an IEEE 802.3az standard that reduces the energy consumption of physical layer (PHY) devices during periods of low link utilization. EEE saves energy by switching part of the transmission circuit to low-power mode when the link is inactive.

An Ethernet link consumes energy even when the link is inactive. EEE provides a method for using energy so that Ethernet links only use it during data transmission. EEE uses a signaling LPI protocol to achieve energy savings when an Ethernet link is inactive. EEE allows PHYs to exchange LPI indications to signal the transition to low-power mode when there is no traffic. LPI indicates when a link can be inactive and when it should resume after a predefined delay, without affecting data transmission. The following copper PHYs are standardized by IEEE 802.3az: 100BASE-T, 1000BASE-T, and 10GBASE-T. However, the optical fiber standards are not.

In other works by [11], algorithms such as the fixed local heuristic threshold (FLHT) and the dynamic local heuristic threshold (DLHT), which are two locally optimized distributed algorithms, can dynamically adjust the number of active sublinks to save energy consumption by approximately 80% in the CN, for both bin packing and load balancing cases. In telecommunications networks with a higher intelligence using SDN controllers, the sleep port algorithm (SPA) and the two-queue algorithm (TQA) are applied, achieving an energy efficiency of up to 50% for grouped links [12,57]. In [13], the algorithm proposed by the author was able to reduce the average number of active links to 25.4%, using a mechanism based on the SMA prediction technique.

Although researchers have proposed many schemes to improve the energy efficiency of the BEs or LAGs, there are still issues with the scalability of the solution, because they are primarily based on a snapshot of the network, without taking into account the scalability and dynamic nature of a TSO network, and most of them are associated with copper standards.

## 3. Traffic Prediction Methodology

The methodology for traffic prediction involves several steps to obtain the best prediction based on the configuration and the model used. The process is illustrated in Figure 1. The diagram starts with raw data input, obtained from an online monitoring platform. The data are then processed for cleaning and adjustments to be used in simulations. Next, the data are divided for validation processes. Then, simulations are run using different neural networks, and their performance is evaluated using metrics such as prediction performance in terms of RMSE, MAE, MAPE, and computational time. Finally, the best prediction model is chosen.

The following section provides more detail on each of these steps.

### 3.1. Data Collection

The utilization of a network monitoring platform is necessary in order to continuously monitor the traffic on routers and provide real-time statistics on various time scales, including but not limited to minutes, hours, days, and months. This results in the accumulation of a significant amount of data, which can be categorized as big data and is structured in the form of a time series. The platform stores this information for a minimum duration of 1 year, and the monitoring system updates the data every 5 min, with the precise time span being contingent on the system in use. The collected data serve as inputs for prediction models. It is imperative to note that for this particular use case, the effective monitoring of the traffic on TSO network routers can only be achieved through the utilization of the network performance monitor platform in conjunction with the simple network management protocol (SNMP).

### 3.2. Structure of RNN, LSTM, GRU, and OS-ELM

The basic architectures used by neural networks have been extensively studied and discussed in the literature; for example, ref. [58] discusses RNNs, ref. [23] discusses LSTM, ref. [59] discusses GRU, and refs. [60,61] discuss OS-ELM.

### 3.3. Data Processing for RNN, LSTM, GRU, and OS-ELM

In this stage, the network performance monitor has provided a database that will serve as the data source for training. However, it is crucial to ensure that this database is suitable for the purpose it is intended for, which is searching. To achieve this, the data will be preprocessed and transformed to align with the specifications of the neural network that will be utilized. The neural network model will play a crucial role in determining the final form of the data and shaping it to meet the necessary criteria for successful searching. In essence, the database will be tailored to the requirements of the neural network, ensuring that it can effectively extract meaningful information from the data and deliver accurate results.

#### 3.3.1. Data Processing for RNN, LSTM, and GRU

The transformation of these three steps will be necessary to better process the data in the implementation of RNN, LSTM, and GRU.

1.Transform the data into a supervised learning problem. In the time series problem, the data are modified as follows: The observation at the last time step (t−1) as the input and the observation at the current time step (t) as the output. This represents the single-step sliding window. It is mainly the only variable to compare, so it is a univariate problem [41].2.Time-dependent time series data. The trend can be removed from the observations and then returned to the original prediction scale. A standard way to remove a trend is to differentiate the data.3.Normalize the observations. The default activation function of the RNN, LSTM, and GRU models is the hyperbolic tangent (tanh), which has values between −1 and 1. The observations will be normalized in the same way, that is, between −1 and 1. This regularization helps to avoid corrupting the experimental set with information from the test dataset.

#### 3.3.2. Data Processing for OS-ELM

The transformation of these three steps is necessary to better process the data in the implementation of OS-ELM.

1.Transform the data into a supervised learning problem. In preparing the data, the model is instructed in the same way as recurrent neural networks, that is, the sliding window or prediction step will be one step.2.Activation function. The activation function of the OS-ELM neural network is the sigmoid function [62]. The rectified linear unit (RELU) activation function, also known as ramp function, is tested in [19], with poor results compared with the sigmoid function.3.Normalize the observations. In the OS-ELM model, the recommended scale is to normalize the data by subtracting the mean and dividing by the standard deviation.

### 3.4. Training and Testing Data

Training and testing datasets make up the two sections of the data collection. The model is constructed and validated using the training and testing datasets, respectively. The forward-chaining method is generally used to compare and validate the models. Forward chaining is a technique used in machine learning to evaluate the performance of a predictive model. In this method, the training data are divided into two parts: a training set and a validation set. The model is trained on the training set and then tested on the validation set. The testing is performed in a forward direction, meaning that the model is tested on data that comes after the training data. Forward chaining is particularly useful when working with time series data, where the goal is to predict future values based on past observations. It is a form of cross-validation that ensures the model is not overfitted on the training data and can generalize well to new data [63,64]. Traditional cross-validation is not suitable for time series data due to temporal dependencies and the arbitrary nature of test set selection, among other factors. It is essential to prevent data leakage when partitioning time series data [65]. In conventional cross-validation, the test set selection is often arbitrary, which may result in the test set error being an unreliable estimator of the error in an independent test set. This problem can be addressed using a technique called stacked cross-validation, as described by [66].

For this case, we applied the same testing set for all simulations, so it will be possible to draw conclusions based on variations of the four neural networks and different hyperparameters. In the case of the training set, data will be added to verify if adding more training data improves the error metrics. As the time series data are seasonal, previous days will be added (*t-xdays*), as shown in Figure 2. The amount of data to be added depends on the time interval of the monitoring system.

Thus, three sets of data in different portions for the simulations are obtained, which are *Training*:*Testing*. Therefore, the first set = 50.0%:50.0%, the second set = 66.7%:33.3%, and the third set = 75.0%:25.0%.

### 3.5. Hyperparameters of RNN, LSTM, GRU, and OS-ELM

Hyperparameters in neural networks are the parameters that are set before training a model, unlike the model parameters, which are learned during the training process. These hyperparameters control various aspects of the model’s training, such as the number of neurons in each layer, the learning rate, the type of activation function, the type of optimization algorithm, epochs, and time steps. The optimal values of hyperparameters greatly affect the performance of the neural network, such as its ability to learn from the data and generalize well to new examples. Finding the best hyperparameters for a specific problem is usually done through a process called hyperparameter tuning, where different values are tried and the performance of each set of hyperparameters is evaluated.

#### 3.5.1. RNN, LSTM, and GRU Hyperparameters

These steps explain the fundamental hyperparameters of the RNN, LSTM, and GRU model that will be implemented.

1.Number of neurons: It is the number of hidden layers added to the RNN, LSTM, and GRU cell.2.Epochs: It is the number of times each training dataset will pass through the neural network.3.Time steps: The number of time steps specified determines the number of input variables *x* used to predict the next time step *h*, as shown in Figure 3. In recurrent neural networks, time steps (also known as lags) refer to the number of previous time steps that are used as input to predict the next time step. For example, if the time steps are set to 3, the network will use the previous 3 time steps of the data as input to predict the next time step. The number of time steps can have a significant impact on the performance of the network, as it determines the amount of context that the network has access to when making predictions.4.Adam optimizer: The Adam algorithm [67] is one that combines RMSProp with momentum. To date, there is no algorithm that has superior performance over others in different scenarios [68], so it is recommended to use the optimization algorithm with which the user feels the most comfortable when adjusting the hyperparameters. For running the simulations, the Adam-based optimization algorithm will be configured for RNN, LSTM, and GRU. Ref. [69] indicates that the Adam optimization algorithm has been a very popular optimizer in deep learning networks in recent years.

#### 3.5.2. OS-ELM Hyperparameters

These steps explain the fundamental hyperparameters of the OS-ELM model that will be implemented.

1.Number of neurons: It is the number of hidden layers added to the OS-ELM cell.2.Forgetting factor: The forgetting factor allows the OS-ELM neural network to continuously forget obsolete input data in the training process, in order to reduce its negative effect on subsequent learning. If the forgetting factor equals 1, it means that the OS-ELM neural network does not forget anything. If the forgetting factor is less than 1, it starts to forget data.

### 3.6. Metrics

The root mean square error (RSME), the mean absolute error (MAE), and the mean absolute percentage error (MAPE) metrics are used to evaluate the performance of prediction models. These metrics provide a systematic approach for evaluating the accuracy of a model’s predictions and facilitate the comparison of the performance of different models. Furthermore, it is important to consider computational time, namely how long it will take to process the prediction.

#### 3.6.1. Root Mean Squared Error (RMSE)

RMSE is a commonly used measure of the difference between the predicted and actual values of a model. It is calculated by taking the square root of the mean of the squared differences between the predicted and actual values and is given by Equation (Equation 1):(1)$$\;\mathrm{RMSE}=\sqrt{\frac{1}{n}\sum_{i=1}^{n}{\left(y_{i}-{\hat{y}}_{i}\right)}^{2}},$$
where $y_{i}$ is the actual value, ${\hat{y}}_{i}$ is the predicted value produced by the model, and *n* is the total number of samples.

#### 3.6.2. Mean Absolute Error (MAE)

MAE is a measure of the difference between the predicted and actual values of a model. It is calculated as the average of the absolute differences between the predicted and actual values. Similar to RMSE, the lower the MAE, the better the fit of the model to the data. MAE is commonly used in time series forecasting; it is less sensitive to outliers than RMSE, and it is defined by Equation (Equation 2):(2)$$\;\mathrm{MAE}=\frac{1}{n}\sum_{i=1}^{n}\left|y_{i}-{\hat{y}}_{i}\right|,$$
where $y_{i}$ is the actual value, ${\hat{y}}_{i}$ is the predicted value produced by the model, and *n* is the total number of samples.

#### 3.6.3. Mean Absolute Percentage Error (MAPE)

MAPE is a measure of the difference between the predicted and actual values of a model. It is calculated as the average of the absolute percentage differences between the predicted and actual values. It expresses the error as a percentage of the actual value, which can be useful for comparing the error of models that make predictions for different scales of values, and it is defined by Equation (Equation 3):(3)$$\;\mathrm{MAPE}=\frac{1}{n}\sum_{i=1}^{n}\frac{|y_{i}-{\hat{y}}_{i}|}{max(\epsilon ,|y_{i}|)},$$
where $y_{i}$ is the actual value, ${\hat{y}}_{i}$ is the predicted value produced by the model, *n* is the total number of samples, and ϵ is an arbitrary small but strictly positive number to avoid undefined results when *y* is zero. It is important to note that when the actual value is zero, MAPE is not defined, which could be a limitation of this measure.

For all neural networks, 100 runs are performed and the mean for each metric is obtained, as in the study by [70], which indicated that in optimization problems, which are heuristic, more than 100 tests should be carried out to find a true or optimal value of the solution.

#### 3.6.4. Computational Time

Computational time or lapsed time refers to the amount of time required to perform a specific computation or task on a computer. It can include the time required to input data, process it, and output the results. It is measured in seconds (s). In the context of neural networks, computational time includes the time required to train and test the network, as well as any other computations that are necessary as part of the model’s implementation.

The hardware and software that will be used in the simulations regarding the neural networks are specified in Table 1.

## 4. Bundle Ethernet Energy Efficiency Methodology

To understand the methodology for energy efficiency in bundle Ethernet, Figure 4 shows how the three cases will be compared. The first case corresponds to the base (without energy efficiency), which will be when the system is in current conditions, that is, how the system is currently functioning. The second and third case will be the development of algorithms that will be responsible for turning on or turning off the corresponding ports on the side that provides services to the router, in order to reduce the energy consumption of the network while meeting traffic demand.

The first algorithm will be based on the past time of the raw values (rv at t−1) of the network performance monitor, in which a threshold or safety factor of 100% will be added, that is, if the traffic at time t−1 was 100 Gbps, a capacity of at least 200 Gbps will be applied to time *t*, and based on that capacity, the required BE will be arranged. This algorithm is called the “threshold-based algorithm”.

The second algorithm will have as input the prediction model that will be selected in the traffic prediction methodology, seen in Figure 1. With this prediction model, the predicted values (*pv at t*) will be obtained for each point in time *t*, with this, the capacity in the BE will be activated. This algorithm is called the prediction-based algorithm.

### 4.1. Threshold-Based and Prediction-Based Algorithms

The logic of both algorithms written in pseudocode are presented below.

#### 4.1.1. Threshold-Based Algorithm

The threshold-based algorithm, which is shown in Algorithm 1, requires the following inputs in order to be executed:$rv_{t-1}$: Raw value of the link speed in Gbps at a previous timestamp, i.e., (rv at t−1). This value is obtained from the Network Performance Monitor platform and is a continuous variable.nl: Number of links initially possessed by the LAG or BE; it is a discrete integer variable and dimensionless.pb: Port bandwidth measured in Gbps, and it is a continuous variable.
**Algorithm 1** Threshold-based Algorithm with raw value in t−1**Require:** raw value in t−1: $rv_{t-1}$; numbers of links in LAG: nl; port bandwidth: pb   $x\leftarrow rv_{t-1}/pb$                  ▹ *x* is defined as a ratio variable   pa←0                        ▹ pa is defined as ports active   pd←nl−pa                   ▹ pd is defined as ports deactivate   **if** 
x>0 
**then**           $pa_{u}=\left(integer\left(x\right)+1\right)*2$      ▹$pa_{u}$ is defined as ports active update           **if** $pa_{u}>nl$ **then**                 pa=nl                 pd=nl−pa                 activate pa ports                ▹ set in router activate ports                 deactivate pd ports             ▹ set in router deactivate ports           **else**                 $pd=nl-pa_{u}$                 $pa=pa_{u}$                 activate pa ports                ▹ set in router activate ports                 deactivate pd ports             ▹ set in router deactivate ports           **end if**     **end if**


As variables that initialize the algorithm, *x* is defined as a ratio between the variables $rv_{t-1}$ and pb. In addition, pa represents the active ports of the link, which is initialized at 0, and it is a discrete-integer variable. pd is defined as the difference between nl and pa. This last variable is obtained from the ports that are deactivated at the time of executing the algorithm.

During the execution of the algorithm, while the variable *x* is greater than 0, meaning the router is present with traffic, the variable $pa_{u}$ is executed, which corresponds to giving a threshold of 100% more than the value obtained from the previous traffic, because the past values are being used for this action. In other words, $pa_{u}$ corresponds to the ports that should be active, according to the previous traffic plus the safety factor defined as double, due to the uncertainty of the future traffic.

After this, if the number of active ports exceeds the number of links of the LAG or BE, defined as nl, it must be limited to the maximum defined by this port channel. Otherwise, it calculates the difference between ports to be activated and deactivated. In both conditions, the number of ports to be activated and deactivated on the router for the next timestamp is executed.

#### 4.1.2. Prediction-Based Algorithm

For this algorithm, the same logic as the previous one (threshold-based) is used; the only and important difference is that it has as input the future traffic value obtained in the prediction. As shown in Algorithm 2, the required inputs are the prediction value pv, the number of links in the LAG nl, and the port bandwidth pb.
**Algorithm 2** Prediction-based algorithm**Require:** prediction value: pv; numbers of links in LAG: nl; port bandwidth: pb   x←pv/pb                       ▹ *x* is defined as a ratio variable   pa←0                         ▹pa is defined as ports active   pd←nl−pa                      ▹pd is defined as ports deactivate   **if** 
x>0 
**then**           $pa_{u}=integer\left(x\right)+1$         ▹ $pa_{u}$ is defined as ports active update           **if** $pa_{u}>nl$ **then**                 pa=nl                 pd=nl−pa                 activate pa ports                ▹ set in router activate ports                 deactivate pd ports             ▹ set in router deactivate ports           **else**                 $pd=nl-pa_{u}$                 $pa=pa_{u}$                 activate pa ports                ▹ set in router activate ports                 deactivate pd ports             ▹ set in router deactivate ports           **end if**   **end if**

For both algorithms to work correctly, the units of measure of the variables pv and pb must be the same.

### 4.2. Metrics

In order to compare both algorithms in addition to the base case, the unit of measure of energy watt-hour (Wh) will be utilized. Watt-hour serves as a means of measuring the amount of generated or performed work. The savings will be reflected in the difference in consumption from the base case per day in relation to each algorithm. It stands to reason that the chosen algorithm will be the one that produces the most savings in comparison with the base case.

The methodology being referred to is a general solution for prediction problems and can be adapted for use in various industries and fields. It involves the use of statistical and machine learning techniques to make accurate predictions about future events. For example, in the energy sector, this methodology can be used to forecast energy demand, anticipate disconnections in power generation centers, and make other predictions that are relevant to the energy industry. Having this information helps energy providers to better manage the supply and demand of energy, which is essential for the efficient operation of their businesses. The use of this methodology in the energy sector can improve the overall reliability of energy services and help to reduce the likelihood of blackouts or other disruptions.

## 5. Case Study

This section describes the network topology to be used in this case study, the traffic description to choose the best neural network in terms of prediction and computational time, and the characteristics of the equipment to be used in order to model the required energy savings as a target.

### 5.1. Network Topology

As previously mentioned, the LAG or BE allow for the logical grouping of multiple Ethernet physical links; this aggregation is treated as a single link and allows for the sum of the nominal speed of each Ethernet physical port used to obtain a high-speed trunk link. In a TSO, different routers can be found connecting to each other. As shown in Figure 5, it can be seen that the LAG connects to *n* optical fiber interfaces generating a LAG of *n* links. Over time, the interfaces of the optical transport network (OTN) and the interfaces of the routers have been growing in the optical IP Core networks of a TSO, reaching up to 400 GE interfaces in other operators [71], thus increasing as traffic demand grows and in accordance with the development of this technology.

In addition, many of the implementations in TSOs are geographically redundant as part of design of reliable communication networks (DRCN) [72], which in this case, as shown in Figure 5, means that Site A is equal to Site B; this means that if the router of Site A goes down (fails), the network traffic will switch to Site B via the interplane link (which connects both sites). The same applies vice versa.

In Figure 6, the architecture of a content distribution network (CDN) is shown. The CDN router aggregates different content providers, such as Google, Netflix, Facebook, Microsoft, and Akamai, among others. The CDN is a group of servers that are distributed geographically and interconnected. They provide cached Internet content from the closest network location to the user to accelerate information delivery.

The Internet output is provided by the router, commonly called the Internet gateway router (IGR). The IGR is the node that aggregates all the traffic of a TSO network and communicates with the Internet. Both the IGR and the CDN router are connected to 8 optical fiber with 100GE interfaces that generate a LAG or BE of 8 links, which is 800 Gbps of capacity. The actual traffic between both routers is approximately 400 Gbps, leaving the same capacity for backup to switch to another site (mirror) in case of failure.

As for the protocol that regulates the hardware providers and provides guidance on the practice of link aggregation for data connections, the IEEE defined the link aggregation control protocol (LACP) within IEEE 802.3ad [73], which is a standard-based method for controlling the aggregation of physical network links. Active LACP mode is the protocol generally configured on equipment to manage aggregated links. This means that the interface is in a permanently active negotiation state. LACP runs on any link that is configured to be in active state. The active port also automatically initiates negotiations with other ports by initiating LACP packets. Static LACP is configured, which increases the interface’s bandwidth and provides reliability. When an Eth-Trunk or BE member link fails or is not active, traffic is automatically distributed to other available links, thereby avoiding traffic interruption. In addition, Eth-Trunk interfaces operating in LACP static mode can implement load balancing. When a BE or LAG is present, all links are active. As a result, energy consumption occurs on all optical links, as the LAG mechanism constantly sends test packets to check if the link or member is active or inactive. On an energy level, it is a waste of energy, because the link capacity should be projected to the network’s peak traffic demand and in other cases to geographically established redundancy.

To model the energy consumption of a router, one must first know the power. In Figure 7, the service side considered in the present study is shown.

According to the study in [5], calculating the equipment consumption mainly consists of a chassis, a number of line cards, and a number of ports. The energy consumption of the chassis and line cards is fixed, regardless of the traffic load, because they will always be active, so the power of a router $P_{r}$ can be expressed by Equation (Equation 4):(4)$$P_{r}=P_{ch}+N_{c}P_{c}+\sum_{i=0}^{n}N_{p_{i}}P_{p_{i}}*fu_{i},$$
where $P_{ch}$ is the power of the chassis, which is the base power of the equipment. $N_{c}$ is the number of line cards, and $P_{c}$ is the power of the line card. $N_{p}$ is the number of ports, $P_{p}$ is the power of the ports, and fu is the port utilization factor. The port utilization factor is the percentage between the used traffic and the port capacity.

When talking about 100GE interfaces, there are standards that establish different technical norms for different purposes. When referring to connections within the CN, the 100GBASE-LR4 standard is the most commonly used [74], as it corresponds to the IEEE 802.3 physical layer specification for 100 Gb/s with 100GBASE-R encoding over four wavelength division multiplexing (WDM) lanes on single-mode fiber, with a reach of at least 10 km [75]. In the 100GBASE-LR4, 100GBASE-SR10, 100GBASE-SR4, and 100GBASE-ER4 standards, energy efficiency is not available [76]. In all of these standards, the common medium through which data are transmitted is optical fiber. In the IEEE 802.3 bm standard, it is indicated that EEE in 100GBASE-LR4 is optional. Additionally, many providers have not yet integrated this norm into their manufacturing. When EEE is not active, Ethernet standards operate at full power all the time, consuming 100% of the energy, regardless of the traffic load [77]. Therefore, for our case, we can assume that the utilization factor will always be equal to 1.

### 5.2. Traffic Description

For the purpose of the study, real traffic of the architecture shown in Figure 6 is used. The dataset represents the activity from 13 November 2021 to 16 November 2021 (4 days), consisting of traffic from the CDN router located in Santiago, Metropolitan Region, Chile (−33.444285499124504, −70.65611679943314). Historical data were captured from the monitoring system, with raw data that contained the time, average incoming traffic, average outgoing traffic, maximum incoming traffic, maximum outgoing traffic, etc. In the dataset, the output traffic peak is the variable used, as it is the maximum traffic that the network can have. In the monitoring system, the minimum data collection time is every 5 min. As a result, the dataset will consist of a traffic variable measured in Gbps every five minutes.

For the purpose of training, and as shown in the methodology, three groups will be divided according to the number of days to train the neural network. As shown in Figure 8, each day corresponds to 288 observations, and each point represents five minutes, with its respective value in Gbps, because this dataset works as a univariate time series. Therefore, the division of training groups is as follows, maintaining the proportion indicated in Figure 2.

First set: 576 observations:-Training observations: 288;-Testing observations: 288.

Second set: 864 observations:-Training observations: 576;-Testing observations: 288.

Third set: 1152 observations:-Training observations: 864;-Testing observations: 288.

As the CN architecture adds traffic and delivers it to various clients through the CDN router, the traffic type is seasonal due to the nighttime usage feature of major CDNs such as Netflix, Facebook, and Google. Seasonal fluctuations in telecommunications traffic are due to changes in consumer behavior, such as increased usage during holidays and winter months. The adoption of 5G and B5G technologies may lead to increased usage and traffic, but it is still unclear how much impact they will have, as they are in their early stages of development and other factors such as infrastructure, regulations, and competition may also influence their adoption and usage. Ultimately, the impact of these technologies on seasonal traffic will depend on consumer behavior and market demand. Other factors such as the availability of infrastructure, government regulations, and competition from other technologies may also play a role.

### 5.3. Equipment Characteristics

In the network architecture shown in Figure 6, there is a configured BE or LAG of 8 links with 100GE interface, typically indicated as 8 × 100GE. According to the LACP protocol, these are configured with the same weight. This means that the load distribution is equal for all links when traffic is assigned, and therefore, in terms of energy consumption, all links are active. The studied equipment is a Huawei brand NE40E-X8A model, with a base configuration that consumes 784 W (typical power at 25 °C), without adding service cards or uplink links. If only the consumption of the BE on one side of the NE40E-x8A equipment is analyzed, as shown in Figure 7, the following configuration and consumption are shown in Table 2.

The base configuration data, as well as the details of each of the cards that make up the equipment, were obtained by the current configuration of the router in the network. Therefore, as all ports being active, the utilization factor ($fu_{i}$) will be equal to 1. The base power is calculated with Equation (Equation 4), as follows:(5)$$P_{r}=784+2290+\sum_{i=1}^{8}36.51=1656\;W,$$

Therefore, the total base power is 1656 W when all ports are active.

## 6. Traffic Forecasting Results in Case Study

The results of both simulations are presented for RNN, LSTM, GRU, and OS-ELM.

### 6.1. Simulation Results of RNN, LSTM, and GRU

Simulations are performed by varying three hyper-parameters: Time-steps (number of inputs) or more commonly known as lags, number of neurons in the hidden layer and epochs. The variations of each hyper-parameter will be as follows:-Time steps (lags): 1, 4, 8, 16, and 32.-Number of neurons: 1, 10, and 50.-Epochs: 1, 10, and 100.

Once each hyperparameter is varied, the results of the RMSE, MAE, MAPE, and computational time metrics will be obtained. Note that the RMSE, MAE, and MAPE metrics are errors, so they should be close to zero. For a better understanding of these results, graphs are made for each metric, showing each neural network in the different training sets or days. Remember that the training days are 1, 2, and 3 days, with 1 day of testing, which is already explained in the previous point.

The tabulation of the results is presented in Appendix A. The best performance values are shown in bold. The criterion was to have at least two metrics with a lower value in the configuration of the hyperparameters lags, number of neurons, and epochs for each group and deep learning neural network. To better interpret the graphs, Table 3 shows the number of hyperparameter configuration indicated on the x-axis. Each hyperparameter has the following description: lags, number of neurons, and epochs). That is, the hyperparameter number 1 has the configuration of lags: 1, number of neurons: 1, epochs: 1, and so on.

The results of the RMSE metric are shown in Figure 9. The x-axis represents the number of hyperparameter configurations, indicated in Table 3. From number 16, the number of epochs changes to 10, and at number 31, to 100. On the y-axis, RMSE is represented. The data are in Mbps, so the range of these errors is from 5.5 to 9.0 Gbps.

It can be observed that in general, the RNN for the epochs at 1 and 10 behaves with a high degree of error compared with the LSTM and GRU neural networks for 1, 2, and 3 days of training. This is because they have a lower number of epochs. That is why, at 100 epochs, the RNN improves, and it is even better than other networks in some cases. Remember that each cycle of backpropagation and forward correction to reduce loss is called an epoch. Backpropagation consists of determining the best input weights and biases to obtain a more accurate result or minimize losses.

Another conclusion from this Figure 9 is that because the RNN neural network does not have a memory effect, and that in 1 day of training it has a lower amount of data, it has worse performance than in 2 and 3 days of training. This is minimized in other neural networks such as LSTM and GRU by having a greater effect on memory than RNN. Furthermore, we can observe that from hyperparameter 40 to 45, in which the number of neurons is set to 50 and the epochs to 100, varying the number of lags makes the LSTM and GRU networks worse compared with the RNN. It can be inferred that many epochs and inputs may be influencing in some way the memory effect of these two networks, which would lead to the impoverishment of these two neural networks due to an excess of data.

Finally, in the RMSE metric, as can be seen, the lowest point is the hyperparameter 33 of the LSTM neural network for 1, 2, and 3 days of training. Within this network, the configuration of the hyperparameter number 33 is 8,1,100, whose value of RMSE closest to zero is on 1 day of training (first group), which can be found in Appendix A.

The MAE error metric is calculated as an average of absolute differences between the target values and the predictions. MAE is a linear score, which means that all individual differences are weighted equally in the average. From the point of view of interpretation, the MAE metric is preferable, since RMSE has the advantage of penalizing larger errors (outlier values) more, so focusing on the upper limit, which means that the RMSE number tends to be increasingly larger than that of MAE as the test sample size increases. In other words, since the data being analyzed represent a time series, it usually does not show outlier values, so it is preferred to plot this metric.

Figure 10 shows the results of the MAE metric for the RNN, LSTM, and GRU neural networks in their 1, 2, and 3 days of training. On the x-axis, the number of hyperparameter configurations is represented, which is already known from the previous graph, and on the y-axis, MAE is represented.

Note that the curves in Figure 10 are similar to the result of the RMSE metric with some small differences. It is emphasized that the RNN after 3 days of training does not have good performance, as evidenced by the RMSE metric. Finally, the closest RMSE error to zero is achieved by the LSTM neural network after 2 days of training (second group) with a hyperparameter configuration (8,1,100) that can be found in Appendix A.

Since MAPE is a further development of the MAE calculation, there is similarity between both metrics. Both are not sensitive to outliers since they use the absolute difference. MAPE is more understandable than MAE for the final user, because the error value is in terms of percentage.

In Figure 11, the results of the MAPE metric for RNN, LSTM, and GRU in their 1, 2, and 3 days of training are shown. On the x-axis, the number of hyperparameter configurations is represented, and on the y-axis, MAPE is represented.

Here, the error is represented in percentage terms, which leads us to believe that we are working with minimal errors close to 1.9%, which translates to the prediction of the three neural networks having good performances. Now, in Figure 11, it can be seen that at 100 epochs is where the three networks make predictions with the least error and that the neural network with the best results continues to be LSTM.

In Figure 12, the results of the computational metric or lapsed time for RNN, LSTM, and GRU in their 1, 2, and 3 days of training are shown. The x-axis represents the configuration number of the hyperparameters, and the y-axis represents the elapsed time.

One of the predominant factors is the amount of time it takes for the neural network to predict the next value. This time must be less than the input time of the monitoring system, which was established at 5 min (300 s). In Figure 12, it can be clearly seen that as the number of epochs increases, they are determinant in terms of computational calculation. Note that these values vary depending on the computer system. Table 1 shows the equipment used. In this case, a GPU is used to accelerate the vectorial calculations of the gradient of each of the neural networks.

In addition to the number of epochs, the number of days of training increases the elapsed time. This is clearly visible where the results of the RMSE, MAE, and MAPE metrics are closer to zero, that is, centered on the value of 100 epochs.

Another point to note is the stability in terms of computational time possessed by the LSTM and GRU neural networks, which does not vary significantly if the number of lags is increased as the RNN does, which increases the time exponentially.

Finally, in the computational time metric, it would be incorrect to choose the value closest to zero, as it must be accompanied by the metrics seen earlier. Now, it is known that computational times greater than 300 s cannot be selected.

Table 4 shows a summary of the results containing the best metrics of the selected neural networks.

The results indicate that the predictions of the LSTM neural network are considerably more accurate than RNN and GRU. If the number of training days that is being given to each neural network is visualized, in RNN, when these days are increased, the computation time increases considerably. In the case of LSTM, there is a deterioration of the RMSE, MAE, and MAPE metrics, and at the same time, an increase in computational time. For GRU, it behaves similarly to LSTM.

On the other hand, it can be observed that the results of the RMSE, MAE, and MAPE metrics in LSTM are more consistent than in the RNN and GRU networks, because they present the same configuration of hyperparameters (8 inputs, 1 hidden layer, and 100 epochs) in different training days. The LSTM and GRU neural networks are surprising, because even with a complex structure on the RNN, they present low computational times when adjusting the hyperparameters to high values. The best combination of the neural network and training days to choose is the LSTM network with 1 day of training with a configuration of 8 lags, 1 neuron, and 100 epochs, as shown in Table 4.

### 6.2. Simulation Results of the OS-ELM Neural Network

For the OS-ELM network, the same method as for RNN, LSTM, and GRU is used, which consists of varying the value of the hyperparameters. Simulations are carried out by varying two parameters: the number of neurons in the hidden layer and forgetting factor. The value that each hyperparameter will take is as follows:-Number of neurons: 10, 110, 210, 310, 410, 510, 610, 710, 810, 910, 1010, 1110, 1210, 1310, 1410, 1510, 1610, 1710, 1810, and 1910.-Forgetting factor: 0.9, 0.95, 0.99, and 1.00.

Once each hyperparameter is varied, the results of the RMSE, MAE, and MAPE metrics and computational time will be obtained. In Figure 13, the results of four graphs of the RMSE metric are shown, varying the forgetting factor. On the x-axis, the number of neurons is represented. On the y-axis, RMSE is represented.

It can be seen that for different training days of the OS-ELM neural network, the RMSE metric can vary between values of 50,000 to 2700, indicating that the behavior of this network is more sensitive to the variation of its hyperparameters compared with RNN, LSTM, and GRU. The tabulation of the results is presented in Appendix A. Remember that if the forgetting factor is 1.00, it means that the OS-ELM network does not forget anything. The forgetting factor allows for continuously forgetting obsolete input data during the training process in order to reduce its negative effect on future learning.

When the forgetting factor is 0.90, the RMSE is highest when 110 neurons are configured. If the forgetting factor is 0.95, the error decreases as the number of hidden nodes increases and converges after 410 neurons. When the forgetting factors are 0.99 and 1.00, the neural network presents a huge error with 10 neurons, which rapidly decreases as the number of neurons increases. Additionally, it can be seen that as the forgetting factor and number of neurons increase, the OS-ELM network does not show much difference for the different days (1, 2, and 3) that are used as input for training data.

In Figure 14, the results of the MAE metric with varying the forgetting factor are shown. The x-axis represents the number of neurons and the y-axis represents MAE.

These results show the same behavior already indicated in the RMSE metric. Remember that MAE is more robust to outliers and does not penalize errors as severely as RMSE. This is why the OS-ELM network trained for 3 days (green bar) is visualized in Figure 14 with a lower error than in Figure 13. This behavior is due to the presence of extreme values.

Figure 15 shows the MAPE metric by varying the forgetting factor. The x-axis represents the number of neurons and the y-axis represents MAPE.

Figure 15 shows the influence of the data input to the OS-ELM network in percentage terms, with the training of 1 day standing out compared with the other days, as it presents the lowest error. Moreover, the convergence of this network is highlighted after 410 neurons with a forgetting factor of 0.95. However, in the OS-ELM network, overfitting can be observed when the number of neurons is increased and the forgetting factor is greater than 0.95. Overfitting is an undesired behavior of neural networks that occurs when the machine learning model provides accurate predictions for the training data but not for new data. If the forgetting factor is close to 1.00, the network may not forget the previous data, and this can cause the network to provide inaccurate predictions when there is a new behavior.

In Figure 16, the results of the computational time metric for the OS-ELM neural network in its 1, 2, and 3 days of training are shown. The x-axis represents the number of neurons and the y-axis represents the lapsed time or computational time.

It is very clear that the computational times at the forgetting factors of 0.90, 0.95, 0.99, and 1.00 increase considerably with a greater number of neurons. This is why the criterion for choosing the best hyperparameter configuration and training days for the OS-ELM network will be based on time and take into account a forgetting factor of 0.95 that allows for possible traffic variations in the network, in order to avoid overfitting, as seen in Figure 15.

Table 5 presents a summary of the best results for each day of OS-ELM network training with a forgetting factor of 0.95 and 410 as the number of neurons in terms of computational time and convergence for that number of neurons.

When comparing the RMSE, MAE, and MAPE metrics, it can be seen that the network with 2 days of training performs the best. With regard to computational times, they are all around 1.2 s, so the final choice is the OS-ELM neural network with 2 days of input.

### 6.3. Final Neural Network Selection

Table 6 shows a comparison of the four best-rated neural networks according to the RMSE, MAE, and MAPE metrics in each of the training groups. For comparative purposes, the worst of the corresponding RNNs is used as a reference. The results are decisive in terms of prediction. Starting with the RMSE metric, the OS-ELM, LSTM, and GRU neural networks surpass the RNN by 26%, 2%, and 1%, respectively. For the MAE metric, the OS-ELM, LSTM, and GRU networks surpass the RNN by 25%, 3%, and 2%, respectively, and for the MAPE measure, the OS-ELM, LSTM, and GRU networks surpass the RNN by 23%, 6%, and 8%, respectively.

Finally, in terms of computational time, the OS-ELM, LSTM, and GRU networks surpass the RNN by factors of 217.6, 7.2, and 3.9 times, respectively.

For the purpose of reviewing the comparison of the two best networks and differentiating their optimization technique, Table 7 shows LSTM, which is based on the best gradient descent optimization network, and OS-ELM, which uses the Moore–Penrose pseudoinverse. It shows the percentages of the RMSE, MAE, and MAPE metrics, using the worst of them as a reference. In terms of prediction, OS-ELM outperforms LSTM by 24% in RMSE, 23% in MAE, and 16% in MAPE. In terms of computational time, OS-ELM is 30.2 times faster than LSTM.

In Figure 17, the two neural networks (OS-ELM and LSTM) indicated in Table 7 are shown. These present the prediction of the traffic load of the CDN router every five minutes in one day. On the x-axis, the bit rate is represented in Gbps, and on the y-axis, the time is indicated in hours, with each point representing 5 min (300 s).

The final choice of the sequential neural network OS-ELM is clear given the results obtained.

## 7. Results of Energy Efficiency Algorithms in the Case Study

The performance of the proposed energy efficiency optimization algorithms are shown below and indicated in Figure 4. These algorithms are compared with the base case, that is, the system operating under current conditions. Remember that the algorithms are responsible for turning on or off the corresponding ports on one side of the CDN router and the link connection to reduce the energy consumption of the network, while satisfying the traffic demand. The first algorithm is based on past traffic with a threshold value of 100%, called the threshold-based algorithm. The second algorithm is based on traffic prediction performed by the neural network selected in the previous chapter, called the prediction-based algorithm.

The results of the simulations are presented in Figure 18, Figure 19, Figure 20 and Figure 21, which were carried out for the test day presented in Figure 8.

In Figure 18, the results of the accumulated savings on the test day are shown for the base case, threshold-based, and prediction-based algorithms. On the x-axis, the time is represented in hours. On the y-axis, the accumulated savings are represented, measured in W/h.

As can be observed in Figure 18, the savings achieved by the prediction-based algorithm is 4829.58 W/h per day, or 4.83 kW/h per day. On the other hand, when considering the threshold-based algorithm, the savings amount to 2.68 kW/h per day. Clearly, there are no energy savings in the base case. The power of the router is 1656 W, as calculated by Equation (Equation 5); thus, the energy consumed is 39.6 kW/h per day. Consequently, the daily savings are 12.2%, only taking into account one side of the equipment (that pertaining to services), the consumption of the chassis, and that of the cards.

Furthermore, the maximum savings that can be achieved by not utilizing any port of the router is 7 kW/h per day, considering 36.5 W of power per port, multiplied by the 8 ports present in the BE. If we compare the savings of prediction (4.83 kW/h per day) to the total consumption of the ports (7 kW/h per day), the savings are almost 70%.

In Figure 19, the number of active ports based on turning on or off during the testing day is shown for the base case, threshold-based algorithm, and prediction-based algorithm. The x-axis represents time in hours, while the y-axis represents the number of active ports.

Regarding the available capacity of the BE, Figure 20 shows that in the base case, there is a high availability of capacity or, in other words, a low utilization; this is due to the network being configured in such a way that it can absorb a site with similar characteristics in case of failure for backup purposes. In the case of the threshold-based algorithm, there is still available capacity in any situation or event. However, in the prediction-based algorithm, this backup capacity is lost to some extent in exchange for the benefit of the energy savings presented in this study.

In this simulation, the prediction-based algorithm exhibits points where traffic is lost due to less accurate predictions, specifically −0.3 Gbps at 12:05 a.m. and −0.79 Gbps at 10:10 PM, as shown in Figure 21.

It can be seen in Figure 19 that for the base case, all ports are active at all times, and for both algorithms, the use of ports begins to decrease in a stepped manner starting at 2:00 a.m., until reaching the lowest point within the range of 3:00 a.m. to 7:00 a.m.; this is due to low traffic usage during nighttime hours. We should note that in the case of the threshold-based algorithm, there is a high variation at one point in the morning not present in the prediction algorithm, which is observed to be more stable.

For the purposes of proportionality, it is considered low, given that it represents approximately 0.2% of total traffic (400 Gbps); however, for a TSO this is critical, as it would result in packet loss at a specific moment on the network, with subsequent attempts at reconnection. A solution to this would be to consider a safety factor in the prediction.

## 8. Conclusions and Future Works

This paper outlines a systematic method for resolving complex problems requiring precise predictions. Utilizing a neural network as the primary tool for prediction enables high accuracy and adaptability to different data types. In addition, the emphasis on energy efficiency emphasizes the significance of reducing energy consumption and discovering ways to optimize resource utilization.

In the case study, a solution was implemented to address the issue of energy efficiency in the data centers of telecommunications service providers. In order to accomplish this, four recurrent and sequential neural networks were compared, allowing predictions to be made every 5 min using sliding windows and hyperparameters of varying values. OS-ELM is the best high-precision network. In terms of prediction, the OS-ELM, LSTM, and GRU networks outperform the RNN by 26%, 2%, and 1% on the RMSE metric; 25%, 3%, and 2% on the MAE metric; and 23%, 6%, and 8% on the MAPE metric, respectively OS-ELM, LSTM, and GRU outperform RNN in terms of computational time by factors of 217.5, 7.2, and 3.9, respectively. For each one, prediction execution times are shorter than the time required for the system to collect data (less than 300 s). The OS execution of ELM’s time for this effect is approximately 1.2 s due to its simple structure and absence of gradient minimization in its search for the optimal solution.

The simulations were applied to real traffic from a telecommunications service provider, which offers a real solution for energy efficiency and energy savings that can be applied not only to the core part but also to the aggregation networks, where there are a large number of BEs or LAGs and significant energy savings can be achieved. Regarding the base case (current conditions), the threshold-based algorithm yielded 6.8% and the prediction-based algorithm yielded 12.2% energy savings per day. It should be noted that only one side of the equipment was considered for energy savings in this simulation (customer or service side). If extrapolated to a large quantity of equipment, it would represent substantial cost savings.

As mentioned previously, the methodology presented in this paper can be expanded and applied to other industries in future research. The approach’s versatility and adaptability make it a promising solution for a wide variety of prediction issues. As technology and data continue to advance, this methodology can be further developed and improved to provide even more accurate predictions and drive innovation across numerous industries. The potential for future applications and its impact in a variety of fields highlight the significance of this methodology and the need for additional research in this field.

The proposed online sequential extreme learning machine (OS-ELM) scheme holds great potential for addressing energy efficiency in telecoms networks as a whole system challenge. For instance, the deterministic, causal, and universality dimensions of the OS-ELM consider the impact of creating inefficiencies elsewhere in the telecoms and/or other systems while driving networks more efficiently. The prediction scheme can provide near-real-time trade-offs to enhance the flexibility of telecoms networks and demands that are susceptible to efficiency measures. This capability of the proposed scheme serves to augment the energy efficiency networks (EENs) enhancements occasioned with architecture and technologies. Consequently, the current OS-ELM approach will be advanced to enable integrated system-level energy efficiency prediction and optimization across communications and energy systems.

The future direction of research will also explore telecoms equivalents of energy ideas around self-generation, storage, flexibility, and demand reduction. For the considered data centers’ case study, our proposed scheme can enable the segmentation of use/users to predict uses that are wasteful, normal, important, and critical. This four-level classification of data centers will consider the temporally and the spatially complex system-level constraints of the EEN of the telecoms system. Holistic, system-level OS-ELM prediction can provide a ubiquitous, seamless, and deeper understanding of the embedded carbon footprints of telecoms network equipment in data centers. This has the potential to predict and utilize feasible EEN solutions via edge computing and Open-RAN equipment. The proposed scheme will be extended by incorporating the EEN trade-offs to encompass end-to-end costs of energy, centralized cloud, end-to-end shared infrastructure, and radio access technologies (5G/6G). This will enable consumer behavior, end-to-end power consumption budget, and user equipment energy use challenges to be predicted for near-real-time EEN optimization for cost-effective data center operation.

## Figures and Tables

**Figure 1 sensors-23-04997-f001:**
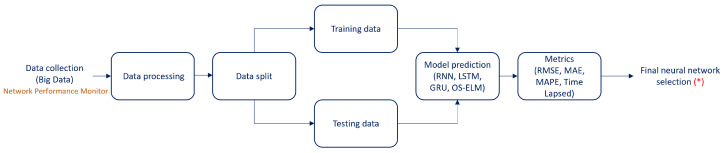
Traffic prediction methodology. * This output will be the input for Bundle Ethernet energy efficiency methodology shown in Section 4.

**Figure 2 sensors-23-04997-f002:**
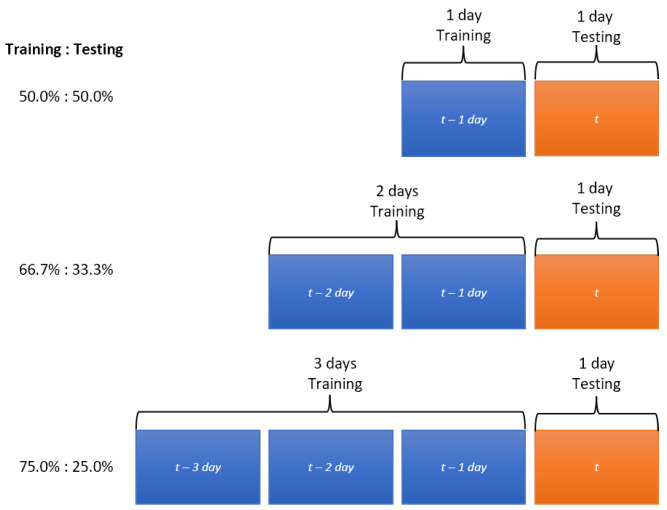
Training and testing dataset.

**Figure 3 sensors-23-04997-f003:**
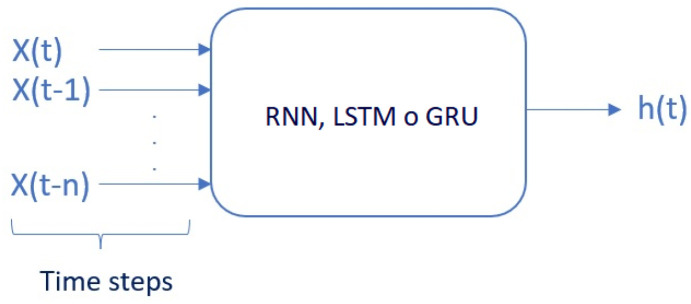
Time steps (lags) in recurrent neural networks.

**Figure 4 sensors-23-04997-f004:**
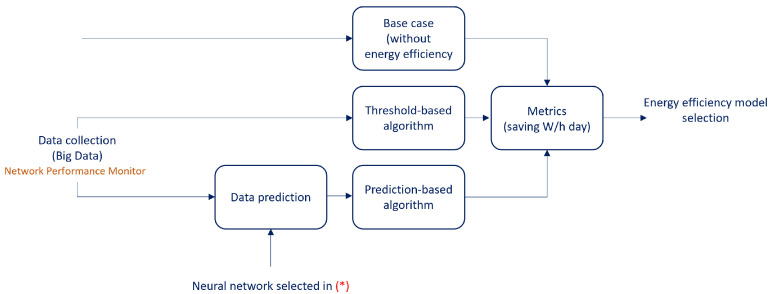
Bundle Ethernet energy efficiency methodology. * This input for Bundle Ethernet energy efficiency methodology is the neural network selected in traffic prediction methodology, according to Figure 1.

**Figure 5 sensors-23-04997-f005:**
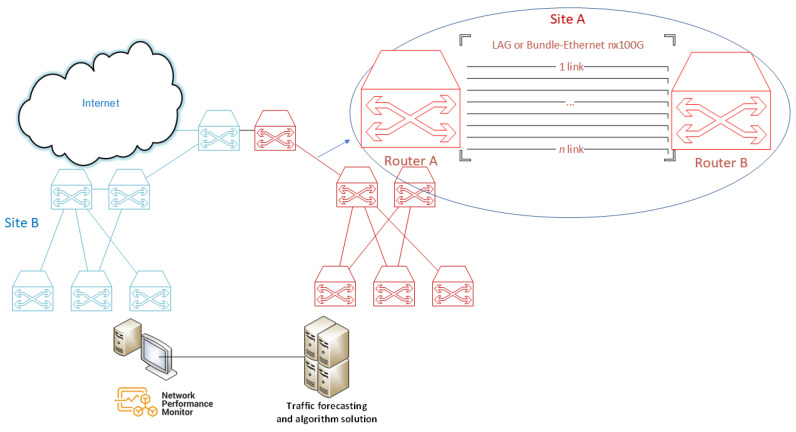
Link aggregation in a TSO.

**Figure 6 sensors-23-04997-f006:**
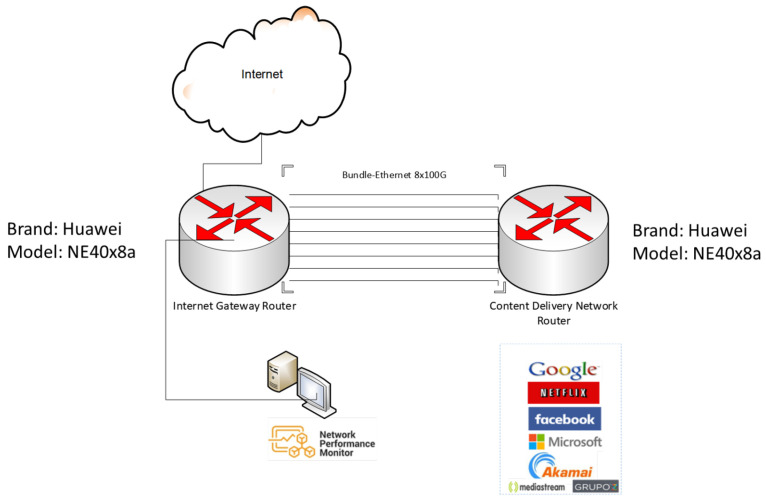
CDN architecture in a TSO.

**Figure 7 sensors-23-04997-f007:**
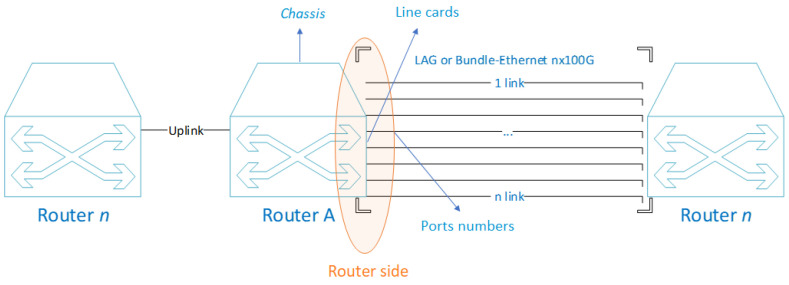
Router side to be considered.

**Figure 8 sensors-23-04997-f008:**
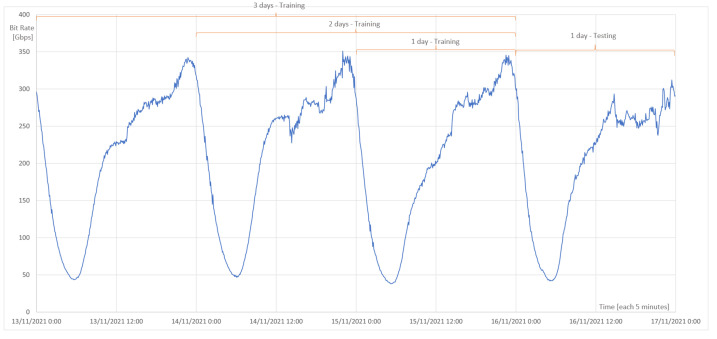
CDN traffic—final dataset.

**Figure 9 sensors-23-04997-f009:**
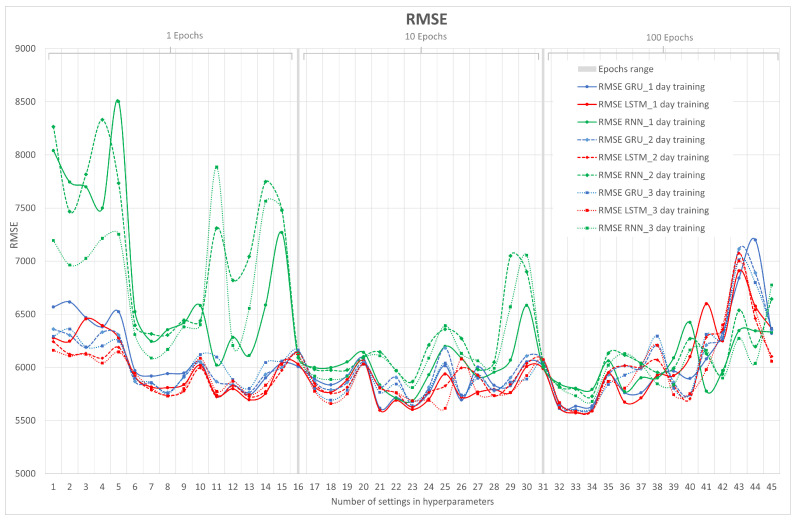
RMSE metrics for RNN, LSTM, and GRU.

**Figure 10 sensors-23-04997-f010:**
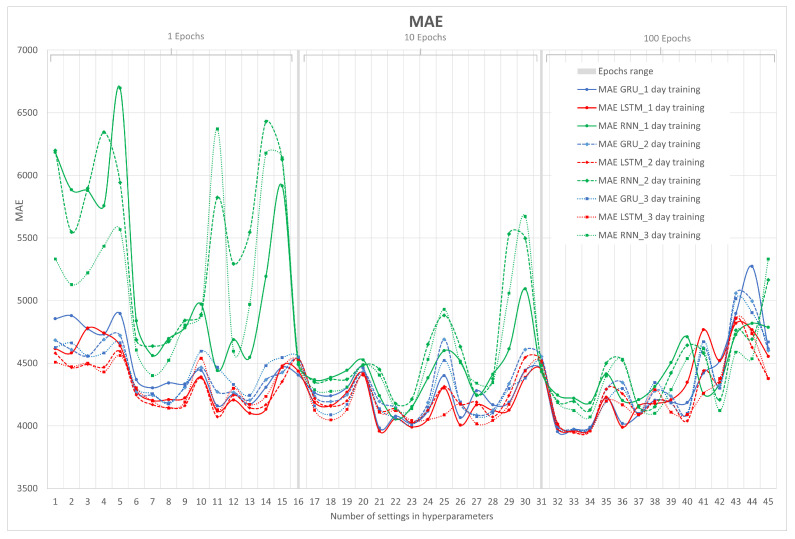
MAE metrics for RNN, LSTM, and GRU.

**Figure 11 sensors-23-04997-f011:**
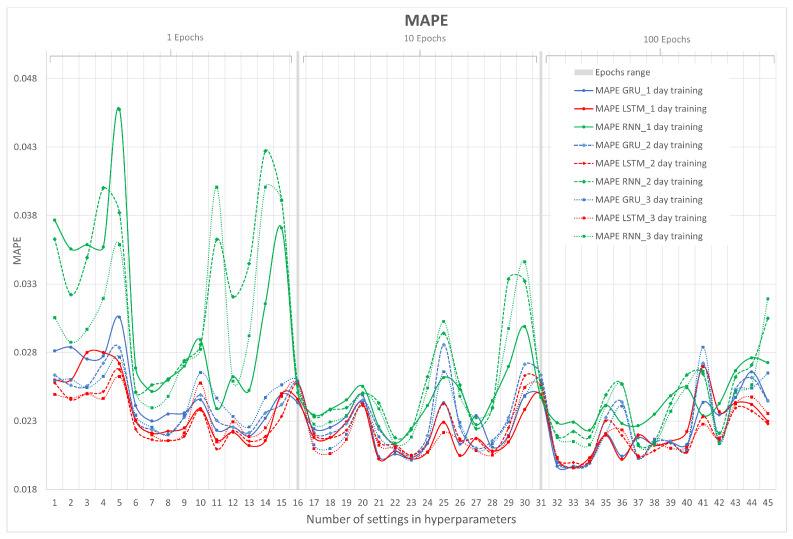
MAPE metrics for RNN, LSTM, and GRU.

**Figure 12 sensors-23-04997-f012:**
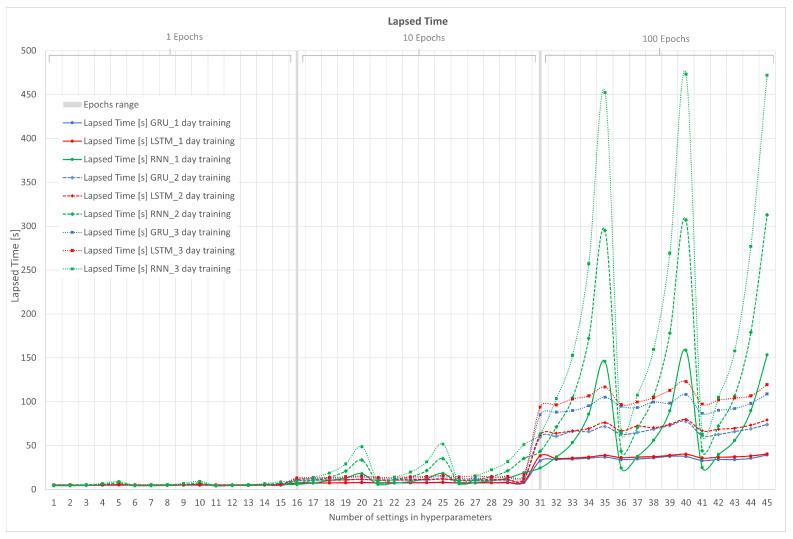
Computational time or lapsed time metric for RNN, LSTM, and GRU.

**Figure 13 sensors-23-04997-f013:**
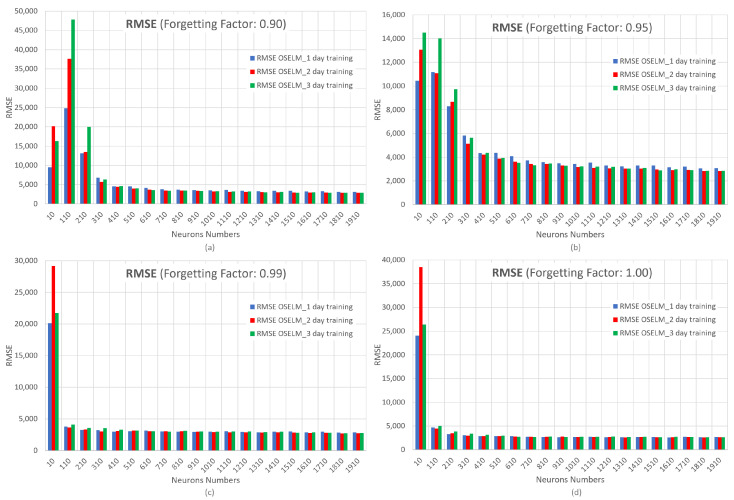
RMSE metric for the OS-ELM network with forgetting factors at (**a**) 0.90, (**b**) 0.95, (**c**) 0.99, and (**d**) 1.00.

**Figure 14 sensors-23-04997-f014:**
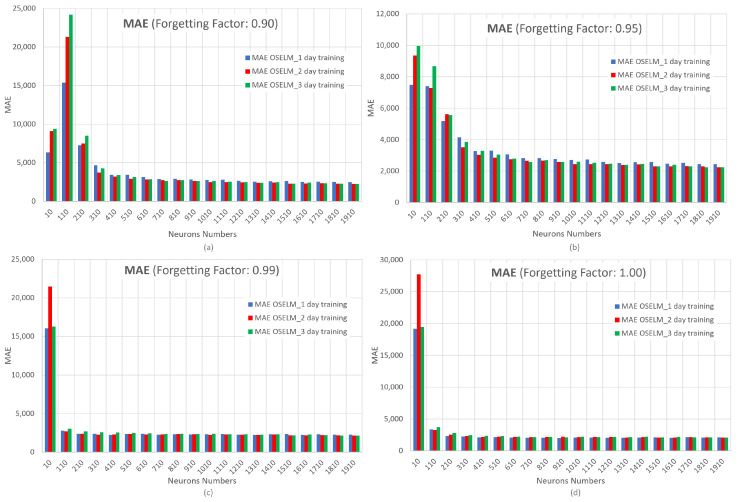
MAE metric for the OS-ELM network with forgetting factors at (**a**) 0.90, (**b**) 0.95, (**c**) 0.99, and (**d**) 1.00.

**Figure 15 sensors-23-04997-f015:**
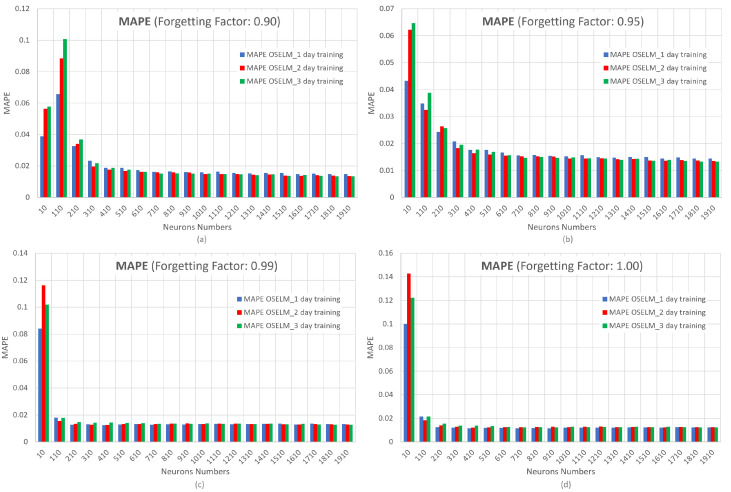
MAPE metric for the OS-ELM network with forgetting factors at (**a**) 0.90, (**b**) 0.95, (**c**) 0.99, and (**d**) 1.00.

**Figure 16 sensors-23-04997-f016:**
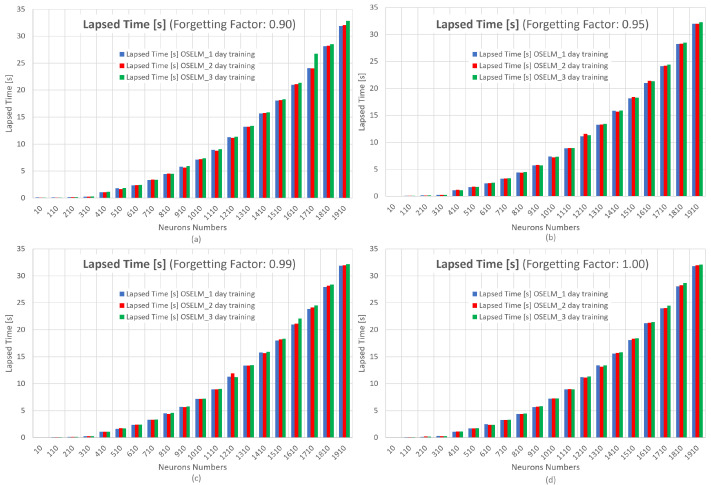
Computational time metric for the OS-ELM network with forgetting factors at (**a**) 0.90, (**b**) 0.95, (**c**) 0.99, and (**d**) 1.00.

**Figure 17 sensors-23-04997-f017:**
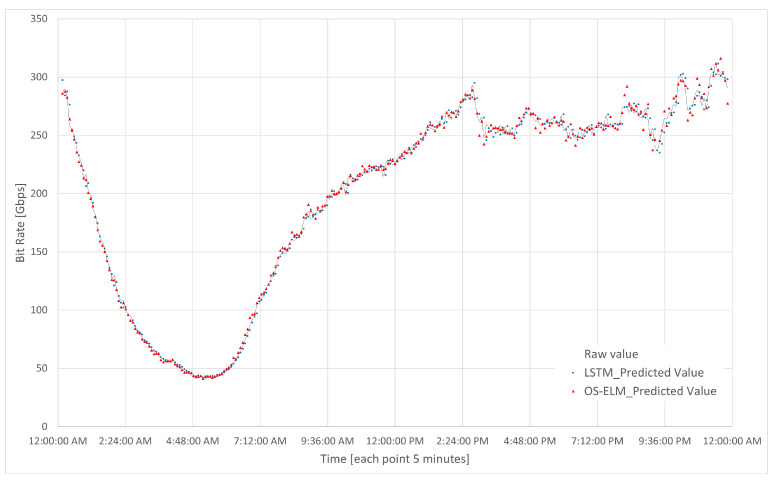
Traffic prediction comparison of the best networks: LSTM and OS-ELM.

**Figure 18 sensors-23-04997-f018:**
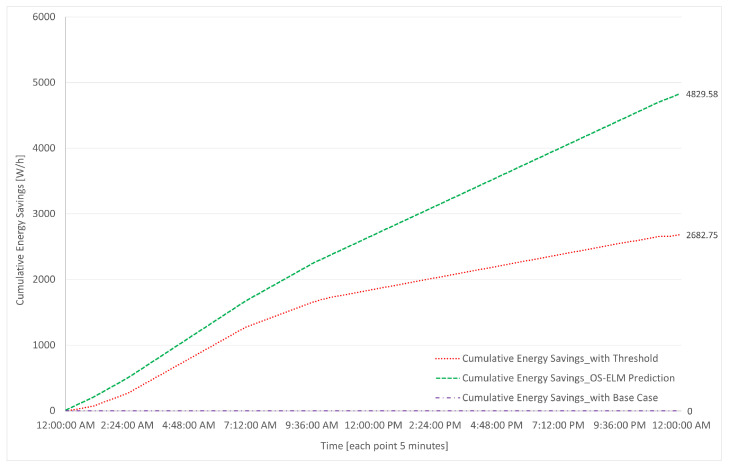
Cumulative one-day savings for base, threshold, and forecast cases.

**Figure 19 sensors-23-04997-f019:**
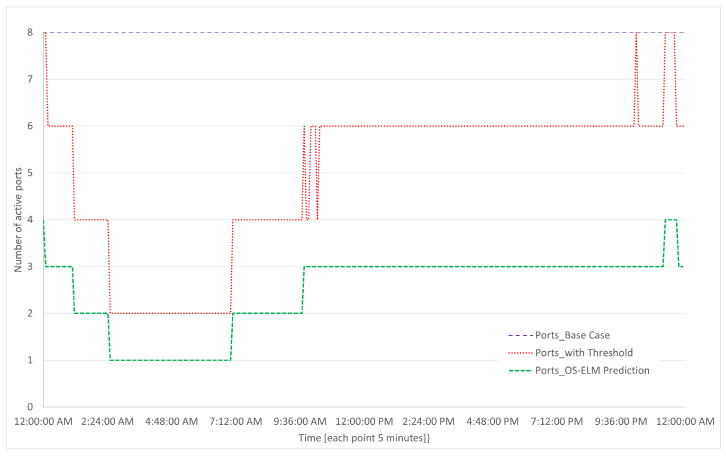
Active number of ports in a day for the base case, threshold, and prediction.

**Figure 20 sensors-23-04997-f020:**
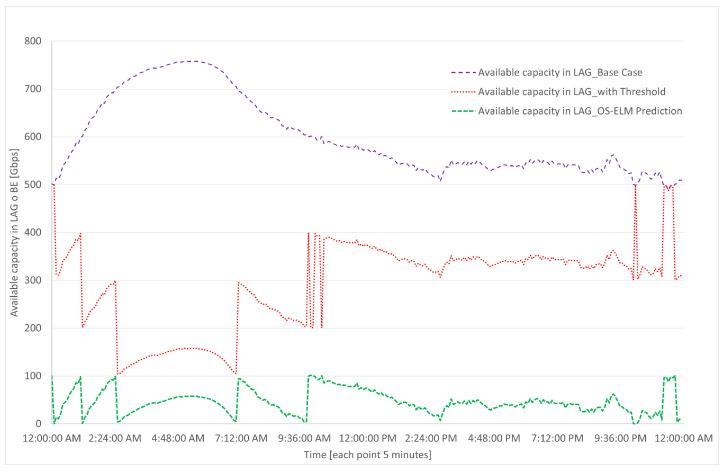
Capacity available in the BE or LAG in a day for base case, threshold, and prediction.

**Figure 21 sensors-23-04997-f021:**
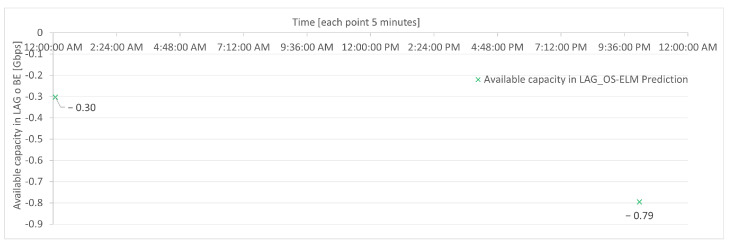
Capacity loss in BE or LAG in one day for base case, threshold, and prediction.

**Table 1 sensors-23-04997-t001:** System specifications.

Hardware	
CPU	Intel(R) Core(TM) 8600 K at 5.1 Ghz
RAM	32 Gb
Graphics card	NVIDIA GeForce(R) RTX 2080.
**Software**	
Python	3.7.10
Tensorflow	2.2.0
Keras	2.3.0
Pandas	1.2.4
Scikit-Learn	0.24.1

**Table 2 sensors-23-04997-t002:** Line card power of CDN router.

Slot	Board Info	Typical Power at 25 °C (W)
Slot1	LPUF-480-E	290
Slot1-PIC0	PIC-2*100GBase-QSFP28	73
Slot1-PIC1	PIC-2*100GBase-QSFP28	73
Slot2	LPUF-480-E	290
Slot2-PIC0	PIC-2*100GBase-QSFP28	73
Slot2-PIC1	PIC-2*100GBase-QSFP28	73

**Table 3 sensors-23-04997-t003:** Hyperparameters configuration.

Number	Hyperparameters Setting	Number	Hyperparameters Setting	Number	Hyperparameters Setting
1	1,1,1	16	1,1,10	31	1,1,100
2	4,1,1	17	4,1,10	32	4,1,100
3	8,1,1	18	8,1,10	33	8,1,100
4	16,1,1	19	16,1,10	34	16,1,100
5	32,1,1	20	32,1,10	35	32,1,100
6	1,10,1	21	1,10,10	36	1,10,100
7	4,10,1	22	4,10,10	37	4,10,100
8	8,10,1	23	8,10,10	38	8,10,100
9	16,10,1	24	16,10,10	39	16,10,100
10	32,10,1	25	32,10,10	40	32,10,100
11	1,50,1	26	1,50,10	41	1,50,100
12	4,50,1	27	4,50,10	42	4,50,100
13	8,50,1	28	8,50,10	43	8,50,100
14	16,50,1	29	16,50,10	44	16,50,100
15	32,50,1	30	32,50,10	45	32,50,100

**Table 4 sensors-23-04997-t004:** Summary of results for RNN, LSTM, and GRU.

Type	Group	Training	Time Step (Lags)	Number Neurons	Epochs	RMSE	MAE	MAPE	Lapsed Time [s]
RNN	First	1 day	4	10	10	5711.161	4054.292	0.02137	7.177
RNN	Second	2 days	16	1	100	5728.889	4126.839	0.02184	172.208
RNN	Third	3 days	16	1	100	5679.286	4070.981	0.02126	257.146
**LSTM**	**First**	**1 day**	**8**	**1**	**100**	**5573.399**	**3961.516**	**0.01959**	**35.745**
LSTM	Second	2 days	8	1	100	5581.399	3947.681	0.01998	66.765
LSTM	Third	3 days	8	1	100	5585.884	3950.845	0.01958	103.195
GRU	First	1 day	4	1	100	5612.749	3953.799	0.01970	34.093
GRU	Second	2 days	8	1	100	5600.504	3972.234	0.01962	66.179
GRU	Third	3 days	8	1	100	5600.595	3971.039	0.01962	89.822

**Table 5 sensors-23-04997-t005:** Summary of results for OS-ELM.

Type	Group	Training	Forgetting Factor	Number Neurons	RMSE	MAE	MAPE	Lapsed Time [s]
OS-ELM	First	1 day	0.95	410	4336.068	3273.084	0.01761	1.128
**OS-ELM**	**Second**	**2 days**	**0.95**	**410**	**4221.912**	**3037.816**	**0.01642**	**1.182**
OS-ELM	Third	3 days	0.95	410	4384.103	3276.136	0.01778	1.113

**Table 6 sensors-23-04997-t006:** Traffic comparison of prediction of neural networks.

Type	Training	RMSE	% RMSE	MAE	% MAE	MAPE	% MAPE	Lapsed Time [s]	Times
OS-ELM	2 days	4221.912	26%	3037.816	25%	0.01642	23%	1.182	217.6x
LSTM	1 day	5573.399	2%	3961.516	3%	0.01999	6%	35.745	7.2x
GRU	2 days	5600.504	1%	3972.234	2%	0.01962	8%	66.179	3.9x
RNN	3 days	5679.286	-	4070.981	-	0.02126	-	257.146	-

**Table 7 sensors-23-04997-t007:** Traffic prediction metrics for the best LSTM and OS-ELM networks.

Type	Training	RMSE	% RMSE	MAE	% MAE	MAPE	% MAPE	Lapsed Time [s]	Times
OS-ELM	2 days	4221.912	24%	3037.816	23%	0.01642	18%	1.182	30.2x
LSTM	1 day	5573.399	-	3961.516	-	0.01999	-	35.745	-

## Data Availability

Not applicable.

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
