# Peer review of "A Novel Traffic Prediction Method Using Machine Learning for Energy Efficiency in Service Provider Networks"

_sensors, 2023, doi:10.3390/s23114997_

Round 1

Reviewer 1 Report

The paper presents a systematic approach for addressing complex prediction problems with a focus on energy efficiency, utilizing neural networks. The authors conducted a case study in the telecommunications industry to improve energy efficiency in data centers. Specifically, they compare four types of neural networks - RNN, LSTM, GRU, and OS-ELM - for the traffic prediction problem in terms of prediction accuracy and computational efficiency. The paper highlights the importance of energy efficiency and the potential for the proposed methodology to be applied to other industries.

The paper provides a thorough and detailed description of the related works, data collection, model structures, data preparation, data splitting, and evaluation metrics used in the study. The authors provide a comprehensive overview of the existing literature on energy-efficient prediction and related works, giving readers important context for understanding the proposed methodology. Moreover, the authors describe the data collection process in detail. They also provide a clear and detailed explanation of the model structures used. The authors also provide a clear explanation of the data splitting process, which is important for evaluating the performance of the models. Finally, the paper includes a detailed description of the evaluation metrics used, including accuracy and computational efficiency, which are crucial for assessing the effectiveness of the proposed approach. The authors also present the experiment results of the compared methods and energy efficiency algorithms in detail.

Major comments:

1.       In addition to the compared methods, it may also be beneficial to compare the proposed approach with other machine learning models, such as tree-based models like XGBoost and LightGBM, as well as the prediction tool developed by Facebook, Prophet. This would provide a more comprehensive evaluation of the proposed methodology and enable readers to better understand its relative strengths and weaknesses compared to other state-of-the-art approaches.

2.       The current experiment setting for computation time may need to be improved by excluding the training time from the computation time metric. Including the training time in the computation time metric can affect the comparison of energy efficiency among the different neural network models. In order to obtain a more accurate comparison of the energy efficiency of the models, the computation time should only include the time required for the models to make predictions on the test data. In addition, it may be more efficient to train a single model using past data over a longer period of time, rather than training a model from scratch every 5 minutes. This approach would allow for a more comprehensive and accurate model to be developed, which could then be tested on the 1-day worth of testing data. By doing so, computational resources could be saved, and the overall efficiency of the model could be improved.

3.       Given the small search space in the case study, it may be more appropriate to use random search for hyperparameter tuning. Random search is a commonly used technique for hyperparameter optimization that can often yield better results than grid search, especially when the search space is large or complex.

4.       The absence of validation data in the current experiment design raises concerns about the possibility of overfitting.

Minor comments:

1.       Hyperparameters such as batch size, dropout rate and so on should also be analyzed.

While the overall quality of English in this paper is good, there are some areas where improvement is needed. Specifically, certain sentences could benefit from rephrasing in order to clarify their meaning and enhance their readability. By making these adjustments, the paper's overall clarity and effectiveness could be further improved.

Author Response

Reviewer 1:

Thanks.

Reviewer 2 Report

The paper proposed a methodology that uses neural networks for energy-efficient prediction. A case study in the telecommunications industry showed that the Online Sequential Extreme Learning Machine (OS-ELM) was the best network in terms of accuracy and computational efficiency, with a potential energy savings of up to 12.2%. The methodology has the potential for application in other industries and is a promising solution for prediction problems. Herein few comments which improve the quality of the article: (1)While the results presented in the paper are satisfactory, we recommend improving the resolution of the figures to enhance their quality and clarity. This will ensure that the data and findings presented in the figures are easily interpretable and provide a strong visual representation of the research. (2)It would be interesting to know more about the potential limitations and future directions of the proposed scheme, as well as how it might be adapted or extended.

The language used in the paper is generally readable, although there are some minor issues with typos and grammar. While these issues do not significantly detract from the overall quality of the paper, addressing them would improve its clarity and professionalism.

Author Response

Dear Reviewer 2:

Thanks.

Round 2

Reviewer 1 Report

Thanks for addressing my comments.